# IgE and Eosinophilia in Newly Arrived Refugees in Denmark: A Cross-Sectional Study of Prevalence and Clinical Management in Primary Care

**DOI:** 10.3390/ijerph22020180

**Published:** 2025-01-28

**Authors:** Kamilla Lanng, Rebecca Vigh Margolinsky, Christian Wejse, Per Kallestrup, Anne Mette Fløe Hvass

**Affiliations:** 1Center for Global Health (GloHAU), Department of Public Health, Aarhus University, 8000 Aarhus C, Denmark; kamillalanng@gmail.com (K.L.); wejse@clin.au.dk (C.W.); per.kallestrup@ph.au.dk (P.K.); annhvass@rm.dk (A.M.F.H.); 2Department of Internal Medicine, Viborg Regional Hospital, 8800 Viborg, Denmark; 3Department of Infectious Diseases, Aarhus University Hospital, 8200 Aarhus N, Denmark; 4Department of Clinical Medicine, Aarhus University, 8200 Aarhus N, Denmark; 5Research Unit for General Practice, Aarhus University, 8000 Aarhus C, Denmark; 6Department of Public Health Programs, Randers Regional Hospital, Central Denmark Region, 8930 Randers NØ, Denmark

**Keywords:** migrant health, refugees’ health, vulnerable populations, eosinophilia, elevated IgE, helminth infections, health assessments, health screenings, primary health care

## Abstract

Refugees have different disease patterns than the population in receiving countries. Furthermore, refugees face barriers to accessing health care services and treatment. The purpose of this study was to describe the prevalence of eosinophilia and elevated IgE levels in refugees and assess the clinical follow-up. Using a cross-sectional study design, we offered health assessments, including eosinophil count and IgE level measurements, to all newly arrived refugees in a Danish municipality from January 2016 to November 2018. In a subgroup, we assessed the clinical follow-up. The study population consisted of 793 refugees, all of whom had eosinophil counts measured, with 411 also having IgE levels measured. Notably, 48.6% were female and most participants originated from Syria, Eritrea, Iran or Afghanistan, with smaller representation from several other countries. Notably, 6.8% had eosinophilia and 32.1% had elevated IgE levels. Syrian origin was associated with a lower prevalence of both biomarkers, and Eritrean origin with a higher prevalence. In a subgroup of 116 participants with abnormal results, general practitioners brought attention to the elevated levels in 50.9% of the cases, and 31.0% of these received a diagnosis related to the findings. In total, 98.3% (114) of patients in the subgroup had contact with their GP following the health assessment. In refugees, eosinophilia and elevated IgE levels are common conditions, and underlying causes are often not diagnosed, potentially leading to inadequate treatment and worse health outcomes.

## 1. Introduction

Refugees are often fleeing from countries afflicted by poverty, conflict, and unstable health systems [1]. Therefore, refugees can be vulnerable regarding their health, and they are found to be burdened with a broad range of communicable and non-communicable diseases upon arrival in Europe [2]. Eosinophilia and elevated IgE are frequent conditions found in refugees and migrants. Eosinophilia has been found in 15–27% [3,4] and elevated IgE has been found in up to 57% [5]. Elevation of these biomarkers can be caused by multiple diseases or conditions such as allergies, asthma, or malignancies. Furthermore, helminth infections are by far the most frequent aetiology in returning travellers and migrants [5,6,7,8]. Individuals with helminth infections can be asymptomatic [9,10,11] and thus unaware of the infection [10]. Lack of diagnosis and treatment of these infections can result in chronic infections with severe long-term morbidity such as anaemia, fibrosis in the hepatic and intestinal systems, bladder neoplasia, and hyperinfection syndromes [9,12,13,14]. Migrants are found to have a lower use of primary health care, when compared to native populations [15], and they face linguistic, socioeconomic, and cultural barriers in encounters with the health care system [16,17]. Healthcare providers describe challenges regarding communication, continuity of care, and confidentiality in delivering health care to refugees in high-income countries, which contributes to inequity within treatment [18,19].

Updated knowledge is needed on the burden of eosinophilia and elevated IgE, in the constantly changing refugee population. Further, there has been a lack of attention to the clinical management of these elevations in primary care in recipient countries, which is why the aim of this study was as follows:

(i) To describe the prevalence of eosinophilia and elevated IgE in newly arrived refugees in Denmark.

(ii) To assess the clinical management of eosinophilia and elevated IgE in newly arrived refugees, in primary care settings in Denmark.

To address these two aims, we conducted a study offering systematic health assessment (SHA) to a complete cohort of all newly arrived refugees and family reunifying arriving in Aarhus, Denmark: The AARHAUS study (AArhus Refugee Health Assessments Using Systematic approaches) [20].

## 2. Materials and Methods

We performed a repeated cross-sectional retrospective study of eosinophilia and elevated IgE in a cohort of newly arrived refugees in Denmark. We collected data from systematic health assessments (SHA) upon arrival and from medical record data from general practitioners (GPs).

### 2.1. Setting

The SHA was performed by a medical doctor at the Department of Social Medicine in Aarhus and included a medical interview, a physical examination, and a blood sample panel. All information from the systematic health assessment was compiled in a comprehensive medical record and passed on to the patient’s GP, who carries the further responsibility for investigation, diagnosis, and treatment of the findings from the SHA. Eosinophil count was systematically included in the blood sample panel from January 2016 and IgE level analyses were included from November 2016.

Since the blood samples were collected as part of a screening process rather than based on vital indications, repeat attempts at difficult venous punctures were performed only after consultation with a medical doctor to assess clinical relevance. In cases where venous punctures were challenging in children, it was not always possible to complete all analyses. Further methodological details are available in previous publications [20].

### 2.2. Study Population

The study population consists of refugees with asylum-seeking backgrounds, refugees arriving through UNHCR, and immigrants who are family reunified with refugees. All individuals in these three groups who obtained residence permits in Aarhus municipality from 2014 to 2018 were included in the AARHAUS study cohort and were eligible for inclusion in this study. They were included in this study if they attended and accepted the invitation for the SHA including the blood sample, during the period from 1st January 2016 to 31st December 2018.

### 2.3. Data Collection

Before conducting the study, power calculations for the main study population were performed. Eosinophilia is found in 4% of all individuals who have undergone blood analysis at Danish hospitals [21]. At the time the study was designed, we hypothesised that eosinophilia would be present in up to 20% of refugees, based on the research available at that time [3,4]. To detect a difference with a 5% significance level and 80% power, we would need to enrol 152 individuals who provided a blood sample, including eosinophil count, as a part of the SHA.

We collected data at two time points and settings. From the systematic health assessment, we collected data on (i) demographics, (ii) eosinophil count, (iii) IgE level, (iv) known diseases associated with eosinophilia and/or elevated IgE, and (v) symptoms associated with eosinophilia, and/or elevated IgE: (a) gastrointestinal: pain, diarrhoea, (b) dermatological: itching, rash (c) respiratory: coughing, dyspnoea, (d) and urological: haematuria, dysuria. Information regarding country of origin, symptoms, and known diseases was based on self-reported data. We accessed all information from the SHA through the Department of Social Medicine’s database.

In a sub-study of the clinical follow-up of abnormal levels, data on participants with eosinophilia and/or elevated IgE were collected from the participants’ assigned GPs, as information on GP affiliation was obtained at the SHAs. Danish GPs do not have a collective database, and data were thus collected by accessing patient records at each individual GP clinic. For practical reasons related to the manual data collection, we were not able to obtain data from all clinics that had participants with eosinophilia or elevated IgE affiliated with them. Thus, participants eligible for the sub-study were identified by contacting all GP clinics with ≥ three patients with eosinophilia and/or elevated IgE to optimise resource use while still capturing the majority of relevant cases.

These data were collected retrospectively in a period defined from their first visit to GP after health assessment and ended when the data were collected between September 2019 and January 2020. This represented a period of 0.5–4 years after participants attended the SHA. To investigate the clinical management in primary care, we collected data on the following 6 outcomes: (i) visited GP: did the patient visit their GP after the SHA, (ii) Attention to elevations: from their GP on the elevated laboratory results, by initiating some degree of investigation, diagnosis, treatment, or by referral to specialist (iii) Parasite testing: PCR testing of faeces for Giardia duodenalis, Cryptosporidium hominis, Cryptosporidium parvum and Entamoeba histolytica or microscopic examination of stool for ova and parasites, (iv) Blood samples: analysed for the biomarker which was elevated at SHA, (v) Diagnosed: with a disease associated with elevations in either eosinophil count or IgE level, (vi) Treatment: received for a disease associated with elevations in either eosinophil count or IgE level.

We collected and stored all data in a database created in the data storage system RedCAP version 14.5.36, (Vanderbilt, Nashville, TN, USA).

### 2.4. Processing of Laboratory Samples

Blood samples were analysed for eosinophil count and IgE level at the Department of Clinical Biochemistry, Aarhus University Hospital. The eosinophil count was analysed using flow cytometry on k-EDTA blood performed on Sysmex XE-2100 haematology analyser (Sysmex Coorporation, Hyogo, Japan). Eosinophilia was classified as ≥0.5 × 10^9^ eos/L, when age > 2 months, which applied to all participants. Plasma-Immunoglobulin E level was analysed using ImmunoCAP on serum performed on Phadia (Thermo Fisher, Uppsala, Sweden). Elevated IgE was classified as ≥15 × 10^3^ IU/L when age < 1 year, ≥100 × 10^3^ IU/L when age ≥ 1 year and <6 year, ≥150 × 10^3^ IU/L when age ≥ 6 year and <10 year, ≥115 × 10^3^ IU/L when age ≥ 10 years.

### 2.5. Analyses and Statistics

All statistical analyses were completed using Stata (StataIC version 16 for Windows). Results were stratified by country and region of origin. Countries and regions with fewer than five participants were categorised as “other” due to GDPR regulations. We classified countries of origin into regions according to the United Nations geoscheme [22]. Prevalence of eosinophilia and elevated IgE were provided at 95% confidence intervals. The six variables describing clinical management at GPs were analysed as dichotomised variables. Polytomous variables were dichotomised, and the bivariate analyses of Fisher’s exact test and Pearson’s Chi-square test were performed, testing for associations between countries/regions and prevalence of elevated laboratory results and for associations between clinical characteristics from SHA and attention to elevated laboratory results from GPs. Similarity in distributions of eosinophil count and IgE level separated by attention to elevations from GPs was tested using the Wilcoxon rank sum test. Logistic regression was performed, investigating odds for elevated IgE, separated by country of origin and adjusted for sex and age. *p*-values below 0.05 were considered statistically significant. We observed no missing data and thus no imputation or handling of missing values was required.

### 2.6. Ethics

The Danish Patient Safety Authority approved data collection from the health assessments and GPs for research activity (file number 3-3013-1926/1, file number 3-3013-2624/1 and file number 31-1522-42). The authorisations from the Danish Patient Safety Authorities allowed us to collect data from GPs without collecting individual consent from all participants. All patient data were anonymised for analysis and information was protected according to the General Data Protection Regulation (2016/679) of the Council of the European Union and the Danish Data Protection Agency conditions for handling sensitive personal data were followed. The project was assessed by “The Central Denmark Region Committee on Health Research Ethics”, which concluded approval was not required.

## 3. Results

Between 2014 and 2018, 1126 regular refugees and 492 family-reunified with refugees obtained a residence permit in Aarhus Municipality, giving a total of 1618 resettling individuals. All refugees and family-reunified to refugees were invited to the voluntary health assessment and 1277 (78.9%) participated including blood sampling. In total, 793 attended in the period from January 2016 where eosinophil count was included, and 411 of those were also tested for elevated IgE from November 2016. A total of 169 individuals were eligible for inclusion in the sub-study. We excluded 39 individuals whose GP had ≤3 patients or who had moved from the Aarhus municipality after the SHA. We were not able to collect data on 14 individuals due to GPs not accepting participation in this study. We ended up with a sub-study population of 116 from whom we collected data on clinical management in primary care. A flow chart of study participants is presented in Figure 1.

### 3.1. Eosinophilia and Elevated IgE

The 793 participants were between 0.4 and 69.7 years of age (median age 22.9) and 48.6% were females. The 793 participants originated from 18 different countries; the majority came from the Western Asia region and Syria was the dominant country of origin. In total, 54 (6.8%) had eosinophilia, varying from 5.1% to 23.1% when divided into regions. The prevalence of eosinophilia was significantly higher among Eritreans (21.1%, *p* < 0.001), when compared to the rest of the population (5.7%) and lower among Syrians (5.4% vs. 9.3%, *p* = 0.040). These associations were consistent when considering the regions of these two countries, with a higher prevalence among refugees originating from Eastern Africa and a lower prevalence among refugees from Western Asia. Elevated IgE was present in 132 (32.1%), varying from 26.7% to 46.4% when divided into regions. Again, the prevalence was significantly higher among Eritreans (47.4% vs. 30.6%, *p* = 0.044) and lower among Syrians (24.6% vs. 38.4%, *p* = 0.003), also applying to the regions of these countries. Performing logistic regression, odds for elevated IgE separated by country were adjusted for sex and age, and none of the estimates changed significantly after adjustment. With logistic regression, it was found that origin from Iran was associated with increased odds for elevated IgE (OR = 2.75), when compared to the rest of the study population, see Appendix A. Combined eosinophilia and elevated IgE were found in 17 (4.1%) and were significantly more frequent in those originating from Eritrea (23.7% vs. 1.1%, *p* < 0.001), see Table 1.

### 3.2. Clinical Management in Primary Care Settings

The sub-study population of 116 individuals with elevations was between 0.4 and 69.7 years of age (median age 18.7) at the time of SHA and 39.7% were female. A total of 114 participants (98.3%) had a documented visit to their GP following the SHA. Of these, 59 participants (50.9%) received attention from their GP regarding eosinophilia and/or elevated IgE, as evidenced by the initiation of further investigations, diagnosis, treatment, or referrals. This represented the greatest loss in the cascade of care (55/114), see Figure 2.

The most frequent tests performed for investigation were blood samples in 17 (14.7%) participants and parasite testing in 20 (17.3%). Notably, 15 of those had PCR testing of faeces for Giardia duodenalis, Cryptosporidium hominis, Cryptosporidium parvum, and Entamoeba histolytica, and another four had a microscopic examination of stools for ova and parasites, and one individual had both tests performed. A total of 36 (31.0%) received a diagnosis related to eosinophilia or elevated IgE and 34 (29.3%) received treatment for these diseases, see Figure 3.

We tested for clinical differences between the participants where GPs brought attention to the elevated levels and those where they did not. IgE levels were statistically significantly higher in the individuals where GPs brought attention to the elevations (*p* = 0.040). There was a significantly larger proportion of symptoms from the gastrointestinal system (50.9% vs. 28.1%, X^2^ = 6.29, *p* = 0.012), respiratory system (30.5% vs. 5.3%, X^2^ = 12.46, *p* = 0.000) and known diseases associated with eosinophilia or elevated IgE (35.6% vs. 8.8%, X^2^ =11.99, *p* = 0.001) in the individuals where GPs brought attention to elevations, see Table 2.

Of the 169 participants with elevated laboratory results, 39 (23.1%) had a known disease or condition at the time of the health assessment which could explain the elevations. At the GPs, 36 (31.0%) were diagnosed 17 of these were diagnostic confirmations of diseases or conditions already found at SHA. Of the diseases found at SHA, 30 (76.9%) were variations of allergies and clinical presentations consistent with allergies. The most prevalent diagnoses at GPs were allergies/asthma in 21 (60.0%), followed by parasitic infections in 7 (20.0%) and others in 7 (20.0%).

## 4. Discussion

We present a study on 793 newly arrived refugees and family-reunified with refugees in Denmark; all of these were systematically screened for eosinophilia and 411 of those additionally for elevated IgE. In total, we found 6.8% with eosinophilia and 32.1% with elevated IgE. Our data showed that GPs brought attention to these abnormal findings in half of the cases and that only 31.0% received a diagnosis related to eosinophilia and/or elevated IgE.

We present a study of the prevalence of eosinophilia and elevated IgE from a refugee population with a known denominator. During the study period, the origin of the largest groups of non-EU asylum seekers was similar in Denmark and the rest of Europe [23,24].

Based on these demographic similarities, we assume that our study population is to some extent representative of the overall refugee population in Europe although this cannot be confirmed with certainty. This study is, to the authors’ knowledge, the first of its kind to study the primary care management of eosinophilia and elevated IgE in refugees.

### 4.1. Prevalence of Eosinophilia and Elevated IgE

Elevated IgE was almost five times as frequent as eosinophilia in this study population. This is similar to other studies finding IgE to be more frequent than eosinophilia [5,7]. We found that refugees arriving in Denmark have a higher prevalence of eosinophilia than the Danish background population where eosinophilia is found in 4% of blood samples at GPs [21]. The prevalence of eosinophilia and elevated IgE levels in this study was generally lower than reported in previous research on immigrant and refugee populations. Variability in the composition of refugee populations, influenced by differences in countries of origin, complicates direct comparisons between studies. One study reported eosinophilia in 27% of immigrants originating from Sub-Saharan Africa [6], whereas another study of immigrant children from Sub-Saharan Africa, Northern Africa, and Latin America found eosinophilia in 22.9% and elevated IgE in 56.8% [5]. Barrett et al. [25] described a decreasing prevalence of eosinophilia and parasite infections in migrants and returning travellers to the UK in the period from 2002 to 2015. This correlates with our findings of a lower prevalence of eosinophilia than previous similar studies. Parasitic infections are the most frequent cause of eosinophilia worldwide [26] and several targeted interventions have been carried out to eradicate and reduce parasitic diseases globally [27,28,29,30,31,32]; this could partly explain the reduced prevalence of eosinophilia observed in our study. However, a recent study of unaccompanied refugee minors from 2016 to 2017 in Denmark’s neighbouring country, Germany, found a significantly higher prevalence of eosinophilia of 18.8% [33]. This potentially illustrates an increased vulnerability in exposure to diseases causing eosinophilia in unaccompanied refugee minors, compared to refugees in general.

We found that Syrian and Western Asian origins were associated with a lower prevalence and that Eritrean and Eastern African origins were associated with a higher prevalence of both eosinophilia and elevated IgE. Origin from Sub-Saharan Africa is previously found to be associated with a high prevalence of both eosinophilia and elevated IgE [5,34]. Our findings of lower overall prevalence compared to similar studies are likely partly explained by the low proportion of participants originating from Middle and Eastern Africa relative to Western Asia.

Health screening of refugees worldwide is not standardised and is conducted from varying local guidelines [35]. The findings of this study suggest that refugees from different regions will present with a different spectrum of diseases. Therefore, origin should be an essential factor included in the development of guidelines for health screenings of refugees, to ensure optimal management. This also emphasises the need for ongoing and updated research when the demographic composition of refugee populations changes.

### 4.2. Clinical Management of Eosinophilia and Elevated IgE

Regarding clinical management in primary care, we observed a low degree of attention to eosinophilia and elevated IgE, as well as a low rate of final diagnosis among the participants for whom the GP initiated follow-up on the elevated values. The causes of this are most likely multifactorial. In Norway, immigrants are found to have a lower use of primary health care services compared to natives [15], which, if applied to the population in this presented study, would influence our results. However, we find that almost all (98.3%) participants visit their GP after the SHA; thus, a lack of use of primary care is not likely the cause of our findings. Though we find the participants in this study access primary care, socioeconomic, linguistic, and cultural factors are still considerable barriers [16] influencing the quality of care for refugees and could contribute to the lack of optimal clinical management we report. Further, newly arrived refugees are found to be burdened with a broad range of communicable and non-communicable diseases [2], and a prioritisation among these by GPs could explain the low degree of attention. Also, due to the timespan from SHA to the collection of data from GPs in our study, there is a risk that the eosinophilia or elevated IgE did not persist, which could explain why the diagnostic process at GPs stopped. Lastly, our results are also a potential consequence of a missing awareness of the potential severity of the underlying diseases, and limited knowledge and experience of clinical management of refugees with eosinophilia and elevated IgE among Danish GPs. This could be supported by our findings of more prevalent use of PCR testing of stool for Giardia duodenalis, Cryptosporidium hominis, Cryptosporidium parvum, and Entamoeba histolytica, than a microscopic examination for ova and parasites. When considering the origin of the study participants, the microscopic examination of stool is more relevant and the recommended diagnostic test [36].

We were able to identify gastrointestinal symptoms, respiratory symptoms, known diseases, and high IgE levels as associated factors to attention to elevations from GPs. This suggests that clinical parameters influence the degree of attention towards the elevations. Previous studies have shown similar results, as larger eosinophil counts were associated with being examined for parasitic infections [37]. As presented, parasitic infections are the most frequent cause of eosinophilia worldwide [26] and among these, especially helminthic infections can be asymptomatic [9,10,11]. This raises concern regarding the neglect of helminthic infections in this population when asymptomatic individuals are less likely to receive attention regarding the elevations. Levels of IgE are found to be associated both with the burden of parasitic infections [5] and with worse symptomatology in children with atopic diseases [38]. This could be one of many possible factors explaining our findings that GPs are more likely to bring attention to those with high IgE levels. However, all participants included in this sub-study were already detected with either eosinophilia and/or elevated IgE and should have been investigated further regardless of degree of elevation.

The low degree of diagnosis and treatment for the causes of eosinophilia and elevated IgE we describe is concerning. Potentially, 70% of the study population lives with undetected diseases that can range from harmful eczema to life-threatening malignancies. The consequence on the individual level is that we fail to detect diseases we already have an indication of, which means that we are adding to the disease burden in a population, which is already found to be exposed and vulnerable regarding their health [2]. On a global level, we fail to contribute to the goal of eliminating helminth infections [39]. It is described that the movements of populations are linked to the spread and lack of control of infections [40], including parasitic helminth infections [41]. We found a probability that these infections are underdiagnosed in Denmark [42], regardless of providing SHA, including screening for eosinophilia and elevated IgE to newly arrived refugees. The results of this and other studies [20,43,44] suggest the benefits of implementing SHA for all refugees arriving in Denmark, as it is not the case now. Follow-up on participants with eosinophilia and elevated IgE at SHA should be systematised to optimise clinical management, and the causes of these elevations should always be chased, as also supported by other recent articles [45,46]. Parasitosis should especially be considered and examined when a refugee from an endemic area presents with eosinophilia and elevated IgE and no known cause. This is in accordance with ECDC guidelines, which recommend screening for schistosomiasis and strongyloidiasis in all migrants from high-endemic countries [47].

### 4.3. Study Limitations

This project has several limitations. Firstly, only 78.9% of the refugees participated in the health assessment including blood sampling. There is a risk that the remaining 21.1% may have different results, and those attending the SHA could have a more beneficial health-seeking behaviour. Moreover, not as many were screened for IgE as for eosinophilia. However, the participation rate for this study is still considered high and it includes a substantial number of participants; thus, we believe that the cohort can provide knowledge on the prevalence of eosinophilia and elevated IgE in newly arrived refugees. Secondly, the sub-study population only constituted 68.6% of the total population with elevations. We collected data from the clinics with the highest number of participants, where knowledge and experience in the clinical management of health issues of refugees are likely to be higher. Thus, we may have introduced selection bias and overestimated the clinical management in this study. Thirdly, we chose to describe the prevalence of eosinophilia and elevated IgE separated by origin. The consequences of this were estimated with greater uncertainty. Furthermore, due to the low degree of diagnosis in this study, we were not able to determine the causes of elevations in most of the study population. This poses limitations in comparison to previous studies and prevents us from describing specific diseases that are prevalent or neglected, which could have contributed to optimising the clinical management of eosinophilia and elevated IgE among newly arrived refugees. The logistic regression analysis was only performed on “elevated IgE”, due to the low number of outcomes for “eosinophilia” and “eosinophilia and elevated IgE”. The logistic regression was only adjusted by sex and age. It could have been relevant to adjust for various symptoms and diseases, but these data were not available. Aarhus municipality is one of the few municipalities in Denmark where systematic health assessment is offered to all newly arrived refugees. Therefore, these data are potentially not representative of the general clinical management of eosinophilia and elevated IgE in refugees in Denmark. There is a likelihood that the level of attention, diagnosis, and treatment is found to be higher in the municipalities where SHA is performed, due to the SHA potentially entailing a greater focus on refugee health. Data on success rates of treatment were not available, which could have been a relevant parameter to investigate when evaluating the clinical management. Future studies should investigate this. Lastly, studies like this would ideally have benefited from one control group consisting of native Danes and one of individuals from the participants’ home countries, to be able to thoroughly evaluate the clinical management of these conditions in Denmark. However, due to the participants originating from 19 different countries, this design was not feasible.

## 5. Conclusions

Eosinophilia and elevated IgE are prevalent biomarkers in refugees, and origin from Africa is associated with a high prevalence. The clinical management of these findings among Danish GPs is inadequate and the underlying causes are underdiagnosed. Our findings suggest that we neglect potentially disabling diseases within a population that already faces barriers when interacting with the healthcare system. Focus and attention on the clinical management of refugees with eosinophilia and elevated IgE in recipient countries are needed. This study suggests a need for the development of a systematised approach for the clinical management of refugees with eosinophilia and elevated IgE. More research should provide knowledge on the current causes of eosinophilia and elevated IgE in refugees, the causes of underdiagnosis at GPs, and the success rates of treatments.

## Figures and Tables

**Figure 1 ijerph-22-00180-f001:**
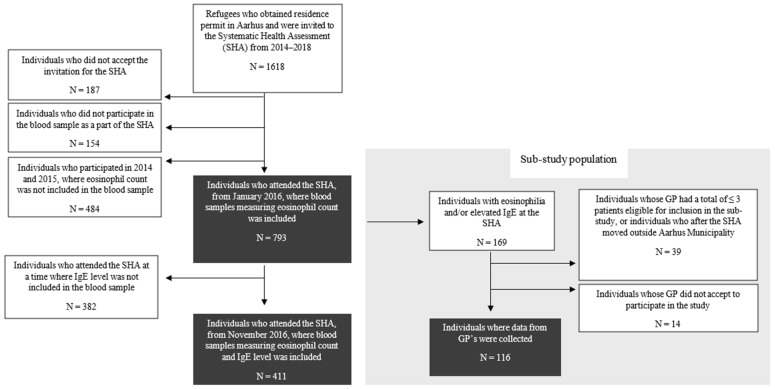
Flow chart of participants.

**Figure 2 ijerph-22-00180-f002:**
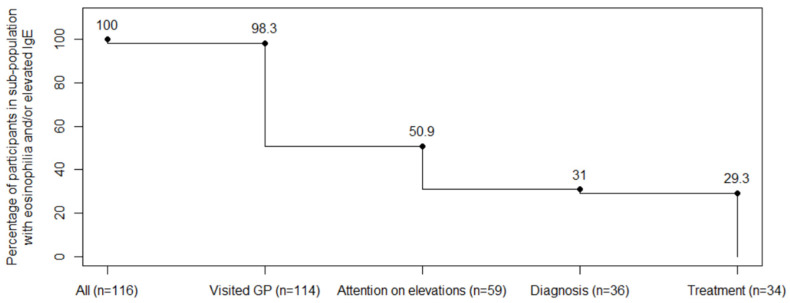
Losses in the cascade of care for eosinophilia and/or elevated IgE.

**Figure 3 ijerph-22-00180-f003:**
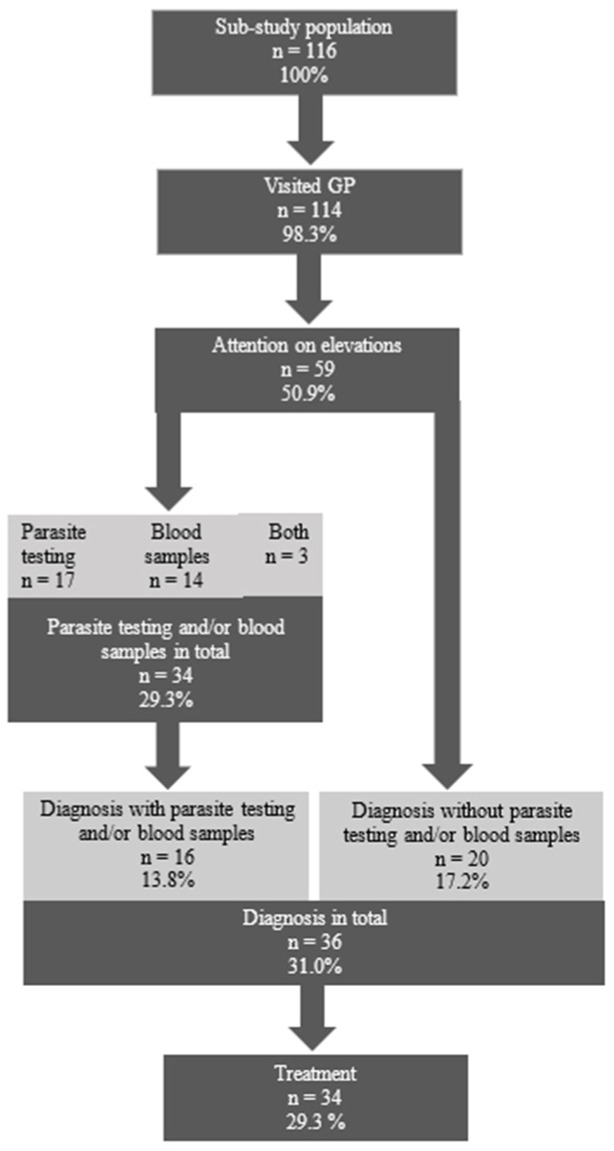
Flowchart of clinical management in primary care provided to patients with eosinophilia and/or elevated IgE.

**Table 1 ijerph-22-00180-t001:** Demographics and prevalence of eosinophilia, elevated IgE, and both biomarkers elevated stratified by country and region of origin.

Origin	Partici- pants	Age Median [Range]	Female	Eosinophilia	Elevated IgE	Eosinophilia and Elevated IgE
Region Country	N	Years	%	n/N (%) [95% CI] *p*-Value	n/N (%) [95% CI] *p*-Value	n/N (%) [95% CI] *p*-Value
**Eastern Africa**	104	18 [1.1–69.7]	54.8	14/104 (13.5) [7.6; 21.6] *p* = 0.004 *	32/69 (46.4) [34.3; 58.8] *p* = 0.005 *	10/69 (14.5) [7.2; 25.0] *p* ≤ 0.001
Eritrea	57	18.9 [1.1–65.6]	56.1	12/57 (21.1) [11.4; 33.9] *p* ≤ 0.001	18/38 (47.4) [31.0; 64.2] *p* = 0.044	9/38 (23.7) [11.4; 40.3] *p* ≤ 0.001
Ethiopia	9	27.2 [6.9–36.9]	66.7	1/9 (11.1) [0.3; 4.8] *p* = 0.472	3/4 (75.0) [19.4; 99.4] *p* = 0.100	0/4 (0) [0.0; 60.2] *p* = 1.000
Somalia	37	16.0 [4.1–69.7]	51.4	1/37 (2.7) [0.1; 14.2] *p* = 0.505	11/27 (40.7) [22.4; 61.2] *p* = 0.394	1/27 (3.7) [0.1; 19.1] *p* = 1.000
Other **	1	19.4 [19.4–19.4]	0.0	0/1 (0) [0; 97.5] *p* = 1.000	0/0	0/0
**Middle Africa**	13	16.5 [3.0–64.2]	46.2	3/13 (23.1) [5.0; 53.8] *p* = 0.052	0/0	0/0
Congo	9	19.1 [5.8–64.2]	44.4	2/9 (22.2) [2.8; 60.0] *p* = 0.120	0/0	0/0
Other **	4	12.2 [3.0–42.0]	50.0	1/4 (25.0) [0.6; 80.6] *p* = 0.246	0/0	0/0
**Western Asia**	531	23.6 [0.4–67.8]	49.5	27/531 (5.1) [3.4; 7.3] *p* = 0.006 *	55/206 (26.7) [20.8; 33.3] *p* = 0.018 *	4/206 (1.94) [0.5; 4.9] *p* = 0.027
Syria	503	23.2 [0.4–67.6]	49.1	27/503 (5.4) [3.6; 7.7] *p* = 0.040	46/187 (24.6) [18.6; 31.4] *p* = 0.003	4/187 (2.1) [0.6; 5.4] *p* = 0.082
Lebanon	8	24.8 [17.7–52.0]	62.5	0/8 (0) [0.0; 36.9] *p* = 1.000	2/5 (40.0) [5.3; 85.3] *p* = 0.658	0/5 (0) [0.0; 5.2] *p* = 1.000
Iraq	8	26.1 [5.5–31.9]	50.0	0/8 (0) [0.0; 36.9] *p* = 1.000	3/6 (50.0) [11.8; 88.2] *p* = 0.391	0/6 (0) [0.0; 45.9] *p* = 1.000
Unknown ***	8	49.9 [26.2–67.8]	62.5	0/8 (0) [0.0; 36.9] *p* = 1.000	4/7 (57.1) [18.4; 90.1] *p* = 0.218	0/7 (0) [0.0; 41.0] *p* = 1.000
Other **	4	49.1 [42.1–56.0]	50.0	0/4 (0) [0.0; 60.2] *p* = 1.000	0/1 (0) [0.0; 97.5] *p* = 1.000	0/1 (0) [0.0; 97.5] *p* = 1.000
**Southern Asia**	138	22.9 [0.4–63.7]	39.9	9/138 (6.5) [3.0; 12.0] *p* = 0.883 *	45/131 (34.4) [26.3; 43.1] *p* = 0.507 *	3/131 (2.3) [0.5; 6.5] *p* = 0.289
Afghanistan	46	17.7 [3.1–57.5]	30.4	2/46 (4.4) [0.5; 14.8] *p* = 0.762	17/45 (37.8) [23.8; 53.5] *p* = 0.401	0/45 (0) [0.0; 7.9] *p* = 0.237
Iran	90	27.0 [0.4–63.7]	44.4	7/90 (7.8) [3.2; 15.4] *p* = 0.658	27/84 (32.1) [22.4; 43.2] *p* = 1.000	3/84 (3.6) [0.7; 10.1] *p* = 1.000
Other **	2	29.3 [16.2–42.4]	50.0	0/2 (0) [0.0; 84.2] *p* = 1.000	1/2 (50.0) [1.3; 98.7] *p* = 0.540	0/2 (0) [0.0; 84.2] *p* = 1.000
**Other ******	7	20.8 [2.4–38.1]	57.1	1/7 (14.3) [0.4; 5.8] *p* = 0.391	0/5 (0) [0.0; 52.2] *p* = 0.181	0/5 (0) [0.0; 52.2] *p* = 1.000
**Total**	793	22.9 [0.4–69.7]	48.6	54/793 (6.8) [5.2; 8.8]	132/411 (32.1) [27.7; 36.9]	17/411 (4.1) [2.4; 6.5]

*p*-values are calculated with Fisher’s exact test unless otherwise specified. * *p*-values are calculated with Person’s Chi-Squared test. ** Countries with less than five participants. *** Participants from border regions or/and areas of conflict, where it is not possible to determine one specific country of origin. **** Regions with less than five participants: Northern Africa, Europe, and South America.

**Table 2 ijerph-22-00180-t002:** Comparison of clinical characteristics between participants in the sub-study whose GP addressed vs. did not address the eosinophilia or elevated IgE.

			All	Attention to Elevations	No Attention to Elevations	*p*-Value
Participants		**n**	116	59	57	
**Elevations**	Isolated eosinophilia	n (%)	24 (20.7)	10 (17.0)	14 (24.6)	0.312
	Isolated IgE		79 (68.1)	40 (67.8)	39 (68.4)	0.942
	Both elevated		13 (11.21)	9 (15.3)	4 (7.0)	0.160
	Eosinophil count, 100 cells/µL *	Median [range]	0.61 [0.5–1.76]	0.66 [0.52–1.76]	0.60 [0.5–1.4]	0.236 **
	IgE level, IU/L *		237.5 [105–1627]	273 [120–1627]	182 [105–928]	0.040 **
**Symptoms**	Gastrointestinal	n (%)	46 (39.7)	30 (50.9)	16 (28.1)	0.012
	Dermatological		26 (22.4)	13 (22.0)	13 (22.8)	0.920
	Respiratory		21 (18.1)	18 (30.5)	3 (5.3)	0.000
	Urological		18 (15.5)	9 (15.3)	9 (15.8)	0.937
Known disease related to eosinophilia or elevated IgE ***			26 (22.4)	21 (35.6)	5 (8.8)	0.001

* Median levels for those with elevated levels of the biomarker. ** Similarity of distribution tested with Wilcoxon rank sum test. *** Known diseases including allergies, asthma, parasites, eczema, and others.

## Data Availability

The original contributions presented in this study are included in the article and Appendix A. Further inquiries can be directed to the corresponding author.

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
