# Peer review of "IgE and Eosinophilia in Newly Arrived Refugees in Denmark: A Cross-Sectional Study of Prevalence and Clinical Management in Primary Care"

_ijerph, 2025, doi:10.3390/ijerph22020180_

Round 1

Reviewer 1 Report

Comments and Suggestions for Authors

Dear Authors,

I am glad I had the opportunity to review this study. It is of significant interest because it focuses on a topic which is frequently neglected also by persons (like me) working with these "atypical" patients.

I only ask you some elucidations and make a couple of comments.

POINTS NEEDING TO BE ELUCUDATED

1. The subgroup you collect from the participating GPs is made of 116 persons, as you clearly show in Fig. 1. I do not understand, why you say in the abstract (line 29)  that 114 had contact to their GP.

2. Line 111 - How did you identify "all GP clinics with >= three patients with eosinophilia and/or elevated IgE"?

3. Lines 166-168 - I do not understand "492 family reunified to refugees". Are these families part of the 1277 persons who participated?

4. Lines 186-187 - "These associations ...". I am not sure I understand this phrase

5. Table 4 is a little confusing and the legend should be more explanatory.

COMMENTS

1. It would be interesting to know the geographic composition of the sub-group seen in the GP clinics. Did the differences seen at the moment of SHA remain? Were the Eritreans still the patients with the highest levels of eosinophils and IgE and Syrians those with the lowest?

2. The "Study limitations" are very clearly and thoroughfully explained. I  agree with what you say at the end (lines 383-386), though frankly I deem it almost impossible to have control groups for all the countries of origin.

3. The references to articles published by your group are appropriate and dlearly evidenced, but, even so, I would suggest to reduce them to three or four, if possible

Comments on the Quality of English Language

Though I am not a native English speaker, I feel that the English could be improved to make the text more clearly understandable.

Reviewer 2 Report

Comments and Suggestions for Authors

This is an informative study with clear goals and objectives. The discussion of the results and their implications for practice is particularly strong in regard to healthcare provider behaviors and attention to disease management and care. 

A few items need explanation.

1- The authors indicated from a power analysis that a sample of 152 needed to be enrolled in the study. However, due to not meeting inclusion criteria or some other exclusion circumstances, only 116 were enrolled in the sub-group analysis. Did the power analysis results only apply to the larger sample that was collected or does it apply to the sub-group? please clarify.

2- The authors elected to not include data where  the GP had ≤ 3 patients. Provide reasoning for selecting this number; and how that may have factored in the results presented.

3- Line 362-363: In acknowledging possible selection bias due to their data collection method and criteria of collecting data from the clinics with the highest number of participants, the authors assert that these clinics are likely to have higher knowledge and experience in clinical management of health issues of refugees.  This needs to be supported with research evidence. Does having a high number of persons with a particular medical condition show up in a clinic equate with better  provider expertise? 

4- The authors at times imply representativeness of their findings to the larger population of refugees in Europe by implying their sample to be representative by assuming association [i.e. line 253-254]. Such assumptive statements should be minimal and only be stated with clear data.  
